# Kimura’s Theory of Non-Adaptive Radiation and Peto’s Paradox: A Missing Link?

**DOI:** 10.3390/biology12081140

**Published:** 2023-08-17

**Authors:** John Herrick

**Affiliations:** Independent Researcher, 3, rue des Jeûneurs, 75002 Paris, France; jhenryherrick@yahoo.fr

**Keywords:** karyotype diversity, genome stability, species richness, species evenness, non-adaptive radiation

## Abstract

**Simple Summary:**

Karyotype diversity, or the interspecific variation in the number of chromosomes and their different forms and sizes in a genome, and species richness, or the number of different species in a phylogenetic clade, are correlated and vary substantially across different animal lineages belonging to the same taxonomic groups, such as salamanders or mammals. The reasons why some lineages are species rich and others are species poor, and why karyotype diversity should correlate so closely with species richness, remain unclear. The following examines these two questions in the context of Motoo Kimura’s hypothesis of non-adaptive radiation and Peto’s paradox. Species richness appears to be inversely correlated with genome stability in mammals and possibly salamanders: lineages with more stable karyotypes tend to be less species rich. Karyotype stability and genome stability, in turn, depend on the cellular DNA damage detection and repair system, suggesting that differences in species richness might be, in part, attributable to lineage specific differences in DNA repair fidelity and efficiency.

**Abstract:**

Karyotype diversity reflects genome integrity and stability. A strong correlation between karyotype diversity and species richness, meaning the number of species in a phylogenetic clade, was first reported in mammals over forty years ago: in mammalian phylogenetic clades, the standard deviation of karyotype diversity (KD) closely corresponded to species richness (SR) at the order level. These initial studies, however, did not control for phylogenetic signal, raising the possibility that the correlation was due to phylogenetic relatedness among species in a clade. Accordingly, karyotype diversity trivially reflects species richness simply as a passive consequence of adaptive radiation. A more recent study in mammals controlled for phylogenetic signals and established the correlation as phylogenetically independent, suggesting that species richness cannot, in itself, explain the observed corresponding karyotype diversity. The correlation is, therefore, remarkable because the molecular mechanisms contributing to karyotype diversity are evolutionarily independent of the ecological mechanisms contributing to species richness. Recently, it was shown in salamanders that the two processes generating genome size diversity and species richness were indeed independent and operate in parallel, suggesting a potential non-adaptive, non-causal but biologically meaningful relationship. KD depends on mutational input generating genetic diversity and reflects genome stability, whereas species richness depends on ecological factors and reflects natural selection acting on phenotypic diversity. As mutation and selection operate independently and involve separate and unrelated evolutionary mechanisms—there is no reason a priori to expect such a strong, let alone any, correlation between KD and SR. That such a correlation exists is more consistent with Kimura’s theory of non-adaptive radiation than with ecologically based adaptive theories of macro-evolution, which are not excluded in Kimura’s non-adaptive theory. The following reviews recent evidence in support of Kimura’s proposal, and other findings that contribute to a wider understanding of the molecular mechanisms underlying the process of non-adaptive radiation.

## 1. Introduction: Formulating the Question

Genetic diversity depends on the continuous and apparently random supply of mutations and genome transformations that ultimately result in reproductively incompatible karyotypes (reproductive isolation). The generation of genetic diversity takes place though varying modes of genetic change, including single base-pair point mutations, gene rearrangements/deletions/amplifications, as well as chromosomal and whole-genome duplications [1,2,3]. Various mechanisms are believed to drive these genetic changes, including DNA replication errors, error-prone DNA repair systems (e.g., non-homologous end-joining), transposable element (TE) activity, recombination-generating DNA polymorphisms, and environmental gene toxins causing DNA damage, as well as miscellaneous misrepair/misreplication events (ionizing radiation, heavy metals, the translesion Y-family of DNA polymerases, etc.) [4,5,6].

In contrast, species richness is believed to occur primarily through natural selection acting on phenotypic diversity: species richness is contingent upon existing gene/allele diversity (DNA polymorphisms) in a population and the ecological factors that favor the reproductive fitness of the corresponding phenotype. The ecological processes through which natural selection operates include changes in climatic and edaphic factors, habitat and niche availability, or other factors such as sexual selection, etc. In terms of evolutionary time, the Neo-Darwinian adaptive theory of evolution is based on a changing environment acting positively on a changing phenotype/genotype, which allows for a given random mutation in an individual to proceed to “fixation”, or substitution, at the population level. Selection from standing genetic variation in a population might, therefore, be the “typical response to an environmental shift” [7,8,9,10,11].

The relationship between species richness (diversity within a taxonomic clade) and the underlying molecular diversity among species has long raised the question of whether or not a causal mechanism somehow links these two forms of diversity during the speciation process [7,8]. Does the rates and levels of the input of mutations drive the rate of speciation, or do the rates and levels of environmental changes, or shifts, in ecological factors determine speciation rates? 

The rate at which speciation occurs is expected, according to the Neo-Darwinian theory, to be largely contingent on the rate of niche change and the rate of change in a lineage’s geographic range (habitat diversity in association with geographic isolation of populations of the same species). In other words, do environmental factors constrain and limit rates of speciation, or are rates of speciation constrained and limited by molecular factors in the species’ germlines? In the former case, the underlying genetic and karyotype diversity would simply be a reflection of a clade’s species richness: higher levels of species richness mean, trivially, correspondingly higher levels of karyotype diversity, and, therefore, adaptive evolution would not be mutation limited but limited by selection [7].

The question, put more directly, becomes: do substitution rates depend directly on mutation rates (genetic drift), or do differing substitution rates depend on differing rates of adaptation (differential selection)? Are non-silent site base substitutions in codons, for example, correlated or uncorrelated with silent site base substitutions in genes across species and taxonomic groups? Some evidence suggests that the two types of base substitution, non-synonymous and synonymous, are indeed correlated, but if and how they might be causally correlated remains to be determined [12,13,14].

## 2. The Neutral Theory of Evolution and the Hypothesis of Non-Adaptive Radiation

A central tenet of Motoo Kimura’s Neutral Theory of Evolution is based on the principle that the rate of substitution in a population should equal the rate of mutation in an individual for selectively neutral mutations (mutation/substitution balance) [15]. Kimura qualifies that “selectively neutral” more accurately means “selectively equivalent” mutations: many different mutations can fulfill the same function or maintain the same level of fitness. Consequently, most mutations that do not negatively affect fitness will “do the job equally well in terms of survival and reproduction of the individuals possessing them” [15]. Expressed otherwise, most mutations—those not adversely affecting an individual’s fitness—can be equivalently selected (or have an equivalent likelihood of being positively selected) because they confer the same fitness value on the individual’s survival and reproductive success. Genetic and karyotype diversity thus reduces to a type of “molecular fitness diversity” among those potential mutations that do not negatively affect the performance or functioning of the phenotype.

Kimura’s “four stage hypothesis” (non-adaptive radiation) proposes a potential solution to the above conundrum of whether rates of mutation constrain, or limit, rates of speciation or whether rates of substitution, or selection and environmental change, account for rates of speciation and the corresponding levels of species richness in a clade. Although Kimura did not specifically refer to “rates of speciation,” his proposal does relate to the striking differences of species richness and levels of species evenness between closely related taxonomic groups (clades) in the same lineage. In the mammalian lineage, for example, the taxonomic clade Rodentia comprises some 2500 different species (among the over 5500 defined mammalian species), whereas the Monotreme clade comprises just five species [16,17]. Is this difference explained because Monotremes inhabit different environments, exposing them to higher rates of extinction, or because they have slower rates of molecular evolution that are less adaptive or, conversely, more maladaptive?

### Kimura’s Hypothesis

According to Kimura’s proposal, macro-evolution consists of four stages (Figure 1A,B):(1)Liberation from the preexisting selective constraint;(2)Sudden increase, or boom, of neutral variants under relaxed selection that are then fixed in the population by random genetic drift;(3)Realization of latent selective potential: some of the accumulated neutral mutants become useful at the phenotypic level in a new environment, which the population is then able to exploit;(4)Intergroup competition, as well as individual selection, leads to extensive adaptive evolution creating a radically different taxonomic group adapted to a newly opened ecological niche.

Kimura’s hypothesis would seem to suggest that genetic variation is increased in response to “liberation” from a previous selective constraint that resisted genetic change (negative selection). Genetic variants then accumulate as a result of genetic drift until they encounter a new selective constraint (environmental resistance). Competition between populations results in the elimination of maladaptive variants (competitive exclusion). Adaptive radiations then occur when new niches become available and more accessible due to the new phenotypes/adaptations that emerge during genetic drift and prior to adaptation.

Kimura believed that this proposal specifically addressed—should it turn out to be correct—the criticism that if most genetic changes are genuinely neutral then, by definition, they cannot be concerned with adaptation: if they are truly neutral, the reasoning goes, how then can they possibly contribute significantly to adaptation and speciation? And, therefore, “the neutral theory is biologically not very important [15]”. A later modification of the neutral theory proposed that most mutations are “nearly neutral” but are maladaptive for the most part. When they are not eliminated by negative selection, they can become fixed in a population if the effective population size is small enough for genetic drift to “overwhelm” selection.

Randomly occurring neutral or nearly neutral mutations should, according to the above criticism, show little or no relationship biologically to either speciation rates or to levels of species richness and evenness. The limiting factor in speciation is the environmentally determined substitution rate and not the molecularly determined mutation rate. The extreme species richness and species evenness in Rodentia, for example, must be due to highly fluctuating habitats and niches rather than to elevated mutation rates in the species belonging to that lineage.

In Kimura’s scenario, the question of whether mutational or ecological processes principally drive speciation becomes moot, if not irrelevant. His hypothesis implies that the two different processes must occur in parallel (mutation/substitution balance) and must be mutually contingent on each other in order to explain how the two independent mechanisms interact to contribute to speciation. What might be the mechanisms governing the balance between mutation and substitution in the process of speciation? Kimura’s hypothesis does not specifically address that question. Nor does it address the question of what might be responsible for the differing rates of molecular change observed across closely and distantly related taxa.

## 3. Peto’s Paradox

Peto’s paradox might contribute some insight into answering those questions. Peto’s paradox is the observation that the evolution of a large body size in vertebrates does not apparently incur higher rates of whole-organism DNA damage, mutation, and genetic diseases such as cancer [18,19,20,21]. Assuming that every single cell in an organism has the same mutation rate, multi-cellular long-lived organisms should be at a higher risk for DNA damage and its negative (or positive) consequences since more cells—and hence more DNA—provide a larger target for mutations, whereas longer life spans provide more time to experience mutation-causing DNA damage. Paradoxically, however, vertebrates with large body sizes, for the most part, are even less prone to DNA damage-related diseases such as cancer, and, consequently, they tend to have significantly longer maximum life spans (MLS) compared to smaller body vertebrates.

Peto’s paradox has usually been explained in terms of the relationship between body size and basal metabolic rate (BMR). According to Kleiber’s law, basal metabolic rate decays with body mass; hence, large animals have relatively lower BMR per mass than smaller animals. One resolution of the paradox—among many other hypotheses—has proposed that species with larger body sizes produce less reactive oxygen species (ROS), which damage DNA, proteins, and lipids. Genome size, however, might also play a role across broad taxonomic groups that range from short lived invertebrates, which have among the smallest animal genomes, to long-lived vertebrates, which have, comparatively speaking, substantially larger genomes [22]. Larger genomes result in larger cells that are consequently expected to have lower metabolic rates.

BMR set by body size and genome size is, therefore, a crucial determinant of MLS. Relatively short-lived mice, for example, have high metabolic rates compared to larger mammals that have significantly lower metabolic rates but similarly sized genomes. Large body size within a taxonomic group and larger genome sizes between taxonomic groups thus correlate, in each case with lower metabolic rates and lower levels of ROS: whales have a larger body size compared to mice but similar genome sizes, whereas salamanders have much larger genomes but mostly smaller body sizes compared to mammals. Thus, genome size might be the crucial factor in explaining Peto’s paradox across taxonomic groups, whereas body size might be the crucial factor explaining Peto’s paradox within taxonomic groups. Recent theoretical and experimental analyses, for example, have shown that organisms with larger genomes are more prone to DNA replication fork stalling, and so require enhanced DNA repair systems [23].

## 4. The DNA Damage Response (DDR) System Mediates the Rate of Mutation Input

The mechanisms that maintain genome stability and determine mutation rates have been extensively studied. The DNA Damage Response (DDR) is a complex damage detection and DNA repair system that evolved in eukaryotes in response to DNA damage and the (non-random) invasion of transposable elements and retroviruses into the genome [24,25,26]. The DDR, therefore, governs the rate of mutation in individuals before mutations can spread through the population either by drift or selection [27,28]. In other words, the DDR controls the level of genetic and karyotype diversity in an individual while, as Kimura would argue, genetic drift mainly—but not exclusively—governs the level of genetic diversity in a population: DNA repair fidelity and efficiency limit mutation/substitution rates in populations and species. The Neo-Darwinians would conversely claim that positive selection in a fluctuating environment alone would determine the level of genetic and karyotype diversity observed in a given population or species. In either scenario, the DDR thus mediates, via mutation input, the level of genetic and karyotype diversity within a population.

### 4.1. The Fidelity and Efficiency of the DDR Varies Significantly across Taxonomic Groups

The longer life spans of large body animals can be understood then as a consequence of a more robust DDR in those species. The mechanisms involved in the positive correlation between maximum life span (MLS) and body size (BS) in vertebrates appear to be quite diverse, but they all converge at the genome level, either directly or indirectly, on the DDR in a manner that promotes, or enhances, genome integrity and stability: K-specialists and vertebrates with large bodies and long life spans tend to have more proficient DDRs than do small-body, short-life-span r-specialists and invertebrates both within and among diverse lineages [29].

Elephants, for example, have a more proficient DDR than smaller mammals do. This is attributed to the fact that elephant cells have multiple copies of the tumor suppressor gene p53, the “guardian of the genome” [30,31,32]. TP53 orchestrates a complex network of factors that results in cell cycle arrest (checkpoint activation), DNA repair, senescence, or apoptosis depending on the amount of DNA damage that the cell has experienced. Elephants consequently benefit from a stronger resistance to cancer compared to other mammals because they rely on an enhanced branch of the DDR that reduces DNA damage by eliminating damaged cells. Other studies have revealed that the number of immune system-related genes also increase with increasing body size, thus reducing the risk of cancer in larger animals [33]. Apparently, these genes promote the elimination of cancer cells that have escaped the cell-based tumor suppressor pathways.

More recently, it has been shown that the bowhead whale relies on more accurate and efficient DNA repair systems to preserve genome integrity. The bowhead has a life span of over 200 years, which is 100-fold more than the two-to-three-year life span of the mouse (*Mus musculus*)—a difference of two orders of magnitude. The bowhead whale thus benefits from a long life span because its cells are more proficient at repairing DNA damage. The study identified two proteins (CIRBP and RPA2) that are present at high levels in fibroblasts and are known to increase the fidelity and efficiency of DNA double-strand break (DSB) repair [34].

Other examples of species with a high resistance to cancer include sharks and salamanders [35,36,37]. The mechanisms are unknown for the longevity and low rates of DNA turnover (DNA damage/genome alterations) found in these particular taxonomic groups, but both groups are known to have low levels of intra-specific heterozygosity, inter-specific genetic diversity, substitutions, single nucleotide variants (SNV), karyotype diversity (KD), etc. Among amphibians, for example, the relatively species-poor salamander lineages have lower rates of genetic change compared to the more species-rich frog lineages [38,39].

The same pattern is observed across taxonomic groups belonging to the Mammalia: different rates of genetic change among different phylogenetic groups, including mice, rats, hamsters, and humans, correlate negatively with MLS. The elevated frequency of mutations in mouse was attributed in these studies to the mutagenic DNA translesion polymerase *eta* [40]. Taken together, these observations explain, at the molecular level, why different taxonomic groups might have different levels of genetic and karyotype diversity and might reflect Kimura’s claim that most mutations are indeed neutral or nearly neutral: mutation rates are equivalent, or directly proportional, to substitution rates (mutation/substitution balance). One would expect then that the varying lineage specific karyotype diversity would likewise reflect varying lineage-specific genome integrity and stability: if mutation rates vary between lineages, then karyotype diversity should also vary.

### 4.2. Sirtuin 6 and the Naked Mole Rat (NMR): A More-Proficient DDR Promotes a Longer Life Span Independent of Body Size

Perhaps the most intensively studied example of a species with a very long life span compared to its body mass/size is the naked mole rat [41,42]. The naked mole rat is a eusocial species of rodent, with a relatively small body size compared to other rodents. Again, in this species, the relatively long life span was found to correlate with genome sequence integrity (genome stability) while deviating significantly from the expected correlation between MLS (per mass) and body size [43]. The Capybara, for example, has a body mass of 55,000 g and a life span of only 15 years compared to the NMR with a body mass of just 35 g but a life span of 32 years. In contrast, the house mouse, which has a body mass of about 40 g, lives only about 4 years [43].

In this model organism (NMR), longevity has been shown to correlate with the efficiency of the error-prone Non-Homologous End Joining (NHEJ) and error-free Homologous Recombination (HR) double-strand break (DSB) repair pathways. Interestingly, longevity does not correlate with nucleotide excision repair [43], which mediates point mutations. This observation appears to be consistent with the earlier observation in salamanders that rates of speciation correlate more with karyotype evolution than with rates of evolution in structural genes [44].

The proteins involved in the DSB repair pathways are more highly expressed in the NMR compared to other related species. The enhanced repair efficiency is also related to the chromatin organizer SIRT6, which has been implicated in mediating the DNA repair of both single-strand and double-strand DNA breaks. SIRT6 also plays an important role in maintaining heterochromatin and repressing TE activity, both of which have been shown to be related to aging [45]. Hence, the correlations between cancer rates, the BMR, and MLS are directly related to more proficient DDRs, independent of body mass, at least in the naked mole rat and other long living small-bodied mammals. The exceptional longevity of squirrels, for example, has been associated with a higher level of genome integrity and stability [46].

## 5. Adaptive Evolution: Mutation Limited, Selection Limited, or Both?

The above review of the relationship between DNA damage, body mass, and MLS suggests that selection acts on different populations having different levels of genetic and karyotype diversity, which depend directly on their different levels of DDR proficiency. Early studies have shown that karyotype diversity is correlated with species richness in a taxonomic group or clade. In 1980, Bengtsson measured standard deviations in karyotype diversity (KD) in mammals and found that the standard deviations of KD in the different taxonomic groups corresponded positively with their respective species richness [47]. Based on these observations, Bengtsson hypothesized that the “properties of stable or unstable karyotypes may indicate that the cytological factors of importance are all of a submicroscopic nature”. One might propose that Bengtsson’s submicroscopic cytological factors include the components of the DDR and relate to the lineage varying DDR proficiencies that underlie stable or unstable karyotypes [47].

Other investigators have made similar observations. Bush, for example, examined 225 different vertebrate species in 1977 and found that rates of speciation are correlated with rates of chromosomal evolution, indicating that high KD reflects rapid speciation rates in species rich genera [48]. None of these studies, however, controlled for phylogenetic relatedness, and hence the findings might represent artifacts due to phylogenetic signal. A more recent investigation of Mammalia that did control for phylogenetic signal confirmed and extended the earlier observations: SR is proportional to and positively correlated with KD independently of species, or taxonomic, relatedness (Figure 2; [49]).

Another study controlling for phylogenetic signal revealed a similar correlation between genome size (C-value) variation and species richness in Urodela. C-value variation, which is a proxy variable for KD in salamanders because of the large range in C-values, was found to correlate strongly with SR (Figure 3). Sister families with pronounced differences in C-values had correspondingly pronounced differences in SR. Related findings revealed a close, negative correlation between heterozygosity and C-values in different salamander lineages [50,51]. Whether or not these correlations were due to a causal relationship between genome stability and rates of speciation, to life history traits and effective population sizes, or some other variable(s) remains an open question.

Perhaps more remarkable than the correlation between KD and SR is the correlation found between KD and species evenness observed in the mammalian phylogenetic tree. Species evenness here refers to the abundance of species in a taxonomic group such as genus, family, or order. At the order level, the mammalian phylogenetic tree is highly skewed (Figure 4), whereas the tree is much less skewed at the genus and family levels (not shown). Presumably, the difference in evenness at different taxonomic levels can be attributed to different rates of speciation and/or extinction since different taxonomic levels in a lineage represent older (group) and younger (genus) radiations. Why such a close correlation between KD and SR evenness would exist is unclear if selection is not limited by mutation or genome integrity and stability. A mutation-limited model of adaptation would seem to be the most parsimonious explanation, but such an explanation still remains to be confirmed.

In contrast, path analysis in the salamander study did not reveal a causal relationship between C-value diversity and species richness but instead showed that variations in C-values more likely occurred in parallel with variations in SR [52]. In other words, the two variables (mutation and substitution/selection) are evolving simultaneously in a manner that determines rates of speciation: differing sizes of adaptive radiation might coincide with and might be contingent on differing rates of change in genome size and structure. Moreover, an abrupt transition in C-value size distributions in mammals was found to occur about 65 million years ago at the Cretaceous–Paleogene (K-Pg) boundary (Figure 5). It is commonly believed that the extinction of the dinosaurs resulted in the adaptive radiation of mammals.

Both of these observations would be consistent with Kimura’s hypothesis of non-adaptive radiation: (1) extinction of the dinosaurs “liberated” mammals from a selective constraint; (2) the small effective population sizes of mammalian species at the time of that event led to a “boom” of variants under “relaxed selection” (genetic drift); (3) some of the “neutral variants” became useful in the new unoccupied niches, habitats, and environments (natural selection); and competition (competitive exclusion) between groups resulted in adaptive radiations and created “radically different taxonomic groups”.

## 6. Other Questions

What does it mean that the two speciation variables (mutation and substitution/selection) are coevolving independently but simultaneously? (1) Selection can act only on mutations that improve (or degrade) phenotypes and only at a rate corresponding to the genetic variability already present at the time of an ecological or environmental shift (such as the “sudden” extinction of the dinosaurs 65 million years ago). (2) The larger the genetic diversity in a population, the faster the rate of speciation will be—necessarily, due to positive selection. Likewise, the faster the mutation rate, the larger the level of genetic diversity on which natural selection can act will be—though not necessarily due to the differential effects of negative selection compounded with genetic drift, which can produce a positive, negative, or no correlation at all between SR and genetic/karyotype diversity.

The correlation between species diversity, meaning the number of different species in a community or ecosystem, and genetic diversity has been extensively studied. Genetic diversity within different communities has been shown to have positive, negative, and null relationships with species diversity [53,54,55,56,57]. As the mechanisms governing mutation inputs and substitutions within a lineage are biologically unrelated and occur at different scales (germline vs. organism and population), there seems to be no reason to expect that KD and SR would be so consistently and so uniformly correlated across diverse mammalian lineages a priori (Figure 4). Although that point might still be open to debate, it is nonetheless difficult to understand in statistical terms why any two taxonomic groups in Mammalia having similar SR would have similar, or nearly similar levels of KD, if selection, rather than drift, is the “typical response” to randomly fluctuating environments.

Transposable elements appear to be another factor associated with varying mutation rates and levels of genome stability across different taxonomic groups [58]. TE activation is not only related to aging but also has been associated with adaptive radiations. TE activation, for example, is positively related to speciation in mammals [59]. High rates of TE activity occur in the genome during the aging process, as the DDR is weakened, and repressive heterochromatin is lost [45]. It would seem reasonable then that loss of repressive heterochromatin as a result of an environmental or some other form of stress might also occur during rapid adaptive radiations in response to a weakened, or “relaxed”, DDR [60].

Are the bursts in TE activity that coincide with recent adaptive radiations due to changes in the proficiencies of the DDR either due to genetic drift [61,62] or due to selection for less (or more) active DDRs? Whether or not these transposition events are causes or consequences of speciation remains unclear [63], but TEs probably reach fixation during speciation events via genetic drift rather than natural selection [64], which would be consistent with Kimura’s hypothesis. The epigenome, and heterochromatin in particular, would seem then to play a crucial role during either adaptive or non-adaptive radiations.

Negative selection reduces and limits genetic variability at the population level, and purifying selection is the preponderant form of natural selection. Genetic diversity is expected to increase in growing populations with small effective population sizes, but a small effective population size also implies a low amount of genetic diversity and heterozygosity. On the other hand, elevated mutation rates are maladaptive and result in genome instability, shorter life spans, and, potentially, higher rates of extinction. The balance between mutation and substitution thus seems to be set by MLS, body size, and reproductive rate. Propagule size, for example, correlates positively with genome size [65]. In plants, genome size has been shown to be negatively associated with species richness [66], which might be related to a nucleotypic effect on cell-growth rate either in the germline or during development [67]. Ecological factors such as a subterranean or cave-dwelling habitat, for example, in some rodent and salamander lineages, certainly play important roles in speciation. What mechanisms might mediate these intriguing correlations remain to be more fully investigated.

## 7. Conclusions

This review has examined the evidence in support of a non-causal but biologically meaningful relationship between the vertebrate DDR (rate of mutation input) and adaptive radiations (substitution/selection rates) in the context of Kimura’s non-adaptive radiation hypothesis and Peto’s paradox: a larger supply of mutations due to a correspondingly weaker DDR will result in a wider spectrum of the karyotype diversity on which positive/negative selection can act and, consequently, a wider/narrower spectrum of species richness among different taxonomic groups and lineages.

The issue then is to better elucidate the molecular and physiological origins of the wide and varying adaptive potential evident in the phylogenetic trees of diverse metazoan lineages, both in vertebrates and in invertebrates, in plants and in animals. As the evidence currently stands, the DDR proposal for explaining the correlated levels of KD and SR within and across metazoan lineages remains a hypothesis complementary to Kimura’s hypothesis that non-adaptive radiation constitutes the principal driving force behind macro-evolution.

## Figures and Tables

**Figure 1 biology-12-01140-f001:**
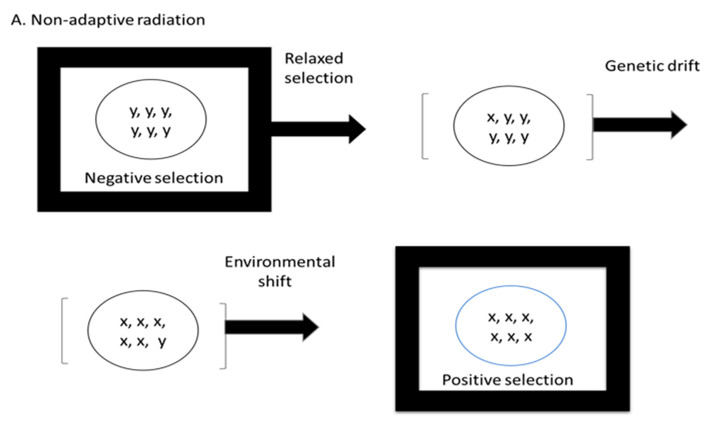
(**A**) Non-adaptive radiation. (**B**) Neo-Darwinian adaptive radiation. Genome size evolution is perhaps the simplest and most studied example of non-adaptive radiation in vertebrates. Genome size increases (or decreases) under the forces of genetic drift. Any increase in genome size, however, must be stabilized for the organism to remain viable. To remain viable over long term macro-evolution, the DNA damage detection and repair system (DDR) must evolve in response to the increase in DNA. The enhanced DDR will stabilize the karyotype and preserve genome integrity. The resulting lower levels of genetic change, for example, in structural genes, will lead to lower rates of speciation in lineages having on average large-genome-sizes.

**Figure 2 biology-12-01140-f002:**
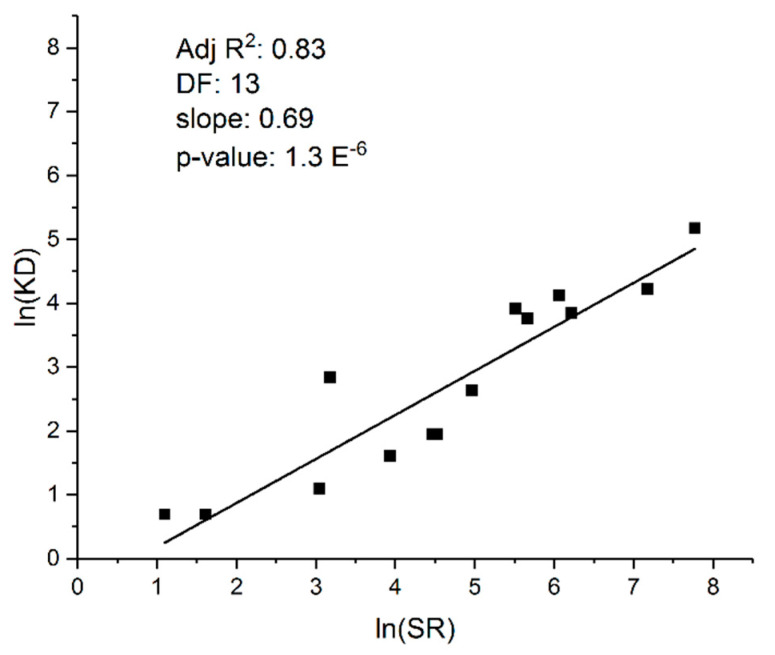
Phylogenetic Least Squares (PGLS) analysis reveals that karyotype diversity (KD) and species richness (SR) in Mammalia are strongly correlated at the taxonomic-order level, thus confirming Bengtsson’s original observation that SR correlates with KD. See reference [49].

**Figure 3 biology-12-01140-f003:**
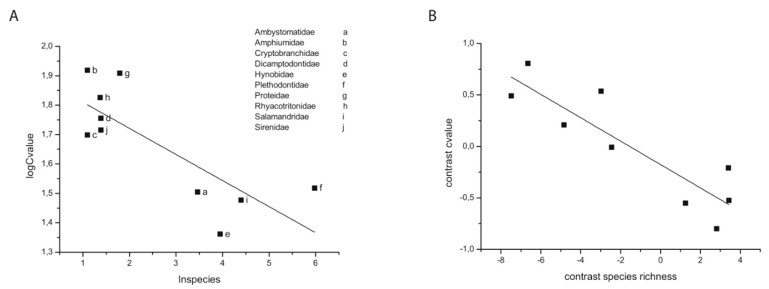
Regression analysis reveals a correlation in salamanders between species richness and genome size (C-value: haploid genome size measured in picograms). (**A**) Ordinary least squares (OLS) regression analysis at the taxonomic family level. The letters refer to the different salamander families examined (inset); (**B**) phylogenetic independent contrasts (PIC) confirm a non-phylogenetically determined relationship between SR and C-value. C-value is considered a proxy variable for karyotype and genome diversity because of the highly variable salamander genome sizes across the Urodela (C-value ranges from 10 to up to 120 pg). Note that controlling for phylogenetic signal substantially improves the correlation, whereas OLS, which does not control for phylogenic signal, reveals two distinct groups depending on life history traits: paedomorphs (Families a, e, f, and i) vs. metamorphic and direct-developing families. See reference [52].

**Figure 4 biology-12-01140-f004:**
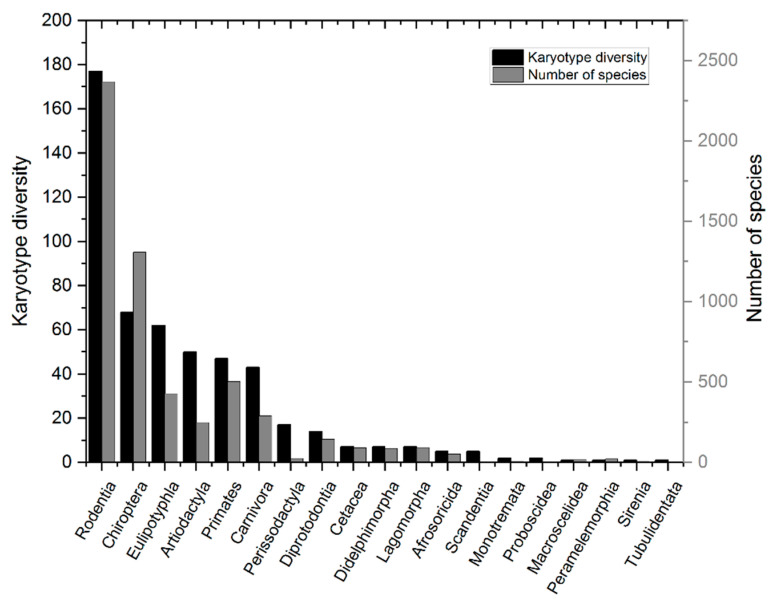
Unevenness in species richness and unevenness in karyotype diversity. Left y-axis: number of different karyotypes per taxonomic order. Right y-axis: number of species per taxonomic order. The findings extend Bengtsson’s original observation revealing a consistency and uniformity of unevenness between species richness and karyotype diversity in the different mammalian orders. See reference [49].

**Figure 5 biology-12-01140-f005:**
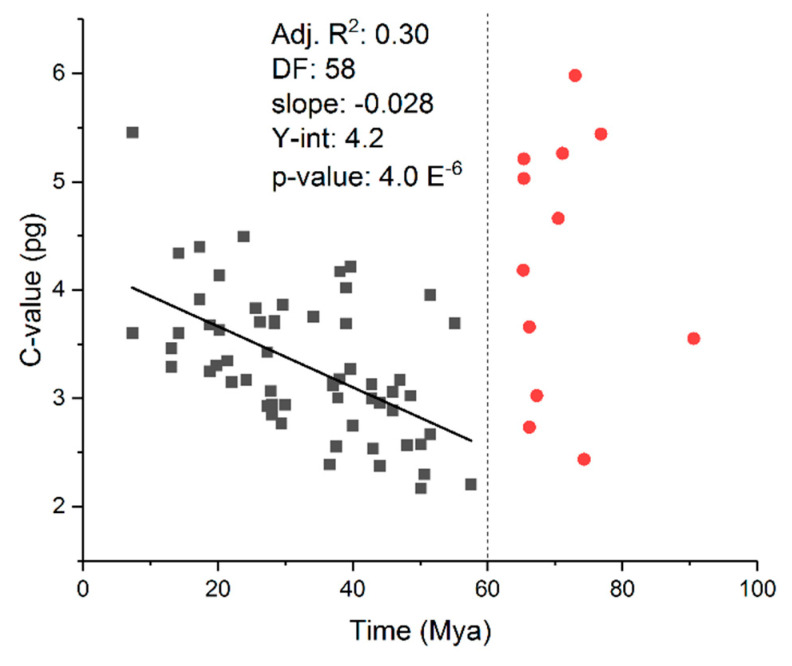
Genome size evolution in Mammalia. OLS analysis was used to regress the clade-stem age (origin in time of the appearance of a taxonomic group in the mammalian Tree of Life) on to clade average C-values at both family and order level clades. Two distinct populations are identifiable: clades with origins before the K-Pg extinction event (red circles to the right of 60 Mya) and clades that emerged after the K-Pg extinction event (black squares to the left of 60 Mya). Note that clades extant before the extinction event have widely varying C-values, and no correlation exists between stem age and C-value. In contrast, clades that emerged after the event exhibit a clear and significant correlation between stem age and C-value revealing that later emerging clades (more recent) have increasingly larger C-values (R^2^ = 0.30; *p* = 4 × 10^−6^; slope = 0.028 pg/Mya; y-intercept = 4.2 pg). Mammalian population sizes presumably increased dramatically after the K-Pg mass extinction due to “liberation” from a previous selective constraint (environmental resistance such as predation, niche availability, etc.) The respective populations then appear to diversify at a fairly constant rate according to a genome size evolutionary “molecular clock”, which might reflect a balance between the forces of genetic drift and natural selection. See reference [49].

## Data Availability

No new data were created or analyzed in this study. Data sharing is not applicable to this article.

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
