# Peer review of "Kimura’s Theory of Non-Adaptive Radiation and Peto’s Paradox: A Missing Link?"

_biology, 2023, doi:10.3390/biology12081140_

Round 1
Reviewer 1 Report
Dear Editor,
thanks for asking me to revise the paper entitled “Kimura’s Theory of Non-adaptive Radiation and Peto’s Para-2 dox: a Missing Link?”. I am afraid to say the paper is not ready to be published in its current form. I list my concerns below.
1. The manuscript is poorly organized. There are some good and clearly described sections, but there is no clear link between them. This is actually apparent from the very beginning of the manuscript. The abstract focuses on the correlation between karyotype diversity and species richness, but then the entire introduction is about genetic diversity and species diversity, and the author gets back to KD and SR in paragraph 5.
2. In this context it is unclear to me whether the author is examining “the relationship between the relationship between KD and SR” (cit. from the abstract) or if the “the distribution in substitution rates is statistically set by the distribution of mutation rate, and therefore causally correlated with it (genetic drift); or is it statistically set by the distribution of rates of adaptation (differential selection), and therefore not causally correlated with mutation rates” (cit. from the introduction). And it is even less clear how he answered one or both questions.
3. The usage of the terms “species diversity” and “species richness” is very misleading. My feeling is that the author is assuming them to be the same thing, but they aren’t. Species diversity is the number of different species in a given community, in a given place and time, influenced by ecological and climatic factors. Species richness (as the author himself states) is the number of species within a clade, which follows a predictable path through time (Foote, 2007) influenced by macroevolutionary processes.
I added some minor comments onto the attached pdf file.
References
Foote, M. (2007). Symmetric waxing and waning of marine invertebrate genera. Paleobiology, 33(4), 517-529.
With very kind regards

English is not bad in itself, word usage is correct. I struggled with sentences construction which was overtly convoluted, with too many subordinate clauses. I'd suggest to revise the manuscript to make it simpler and straighter.
Author Response
Comment 1: Numerous changes have been made to improve the flow of the manuscript and emphasize the underlying connections between seemingly unrelated field such as the molecular biology of DNA repair, aging and DNA damage, Kimura's Non-adaptive radiation hypothesis and speciation.
Comment 2: The sentence has been changed and clarified: “the distribution in substitution rates is statistically set by the distribution of mutation rate, and therefore causally correlated with it (genetic drift); or is it statistically set by the distribution of rates of adaptation (differential selection), and therefore not causally correlated with mutation rates” (cit. from the introduction).
Comment 3: The error has been corrected: "The usage of the terms “species diversity” and “species richness” is very misleading. My feeling is that the author is assuming them to be the same thing, but they aren’t."

Reviewer 2 Report
This review paper takes an in-depth look at the correlation between karyotype diversity (KD) and species richness (SR), using the lens of Motoo Kimura's theory of non-adaptive radiation and Peto's paradox. The paper dissects the relationship between the DNA Damage Response (DDR), adaptive radiations, and species diversity, suggesting a biologically meaningful but non-causal relationship.
This review is commendably comprehensive, weaving together multiple complex concepts such as the DDR's efficiency variations across taxonomic groups, the role of Sirtuin 6 in the Naked Mole Rat's DDR, and Kimura's non-adaptive radiation theory, making the review a robust contribution to the field. Moreover, the paper is well-structured, guiding the reader from an introduction to these concepts, through in-depth examinations of the roles of DDR and adaptive evolution, to a thoughtful conclusion.
One suggestion I would recommend is the inclusion of a comparison diagram to illustrate the differences between adaptive theories of evolution and Kimura's non-adaptive theory. This diagram could serve to organize the various theories and concepts discussed in the paper, demonstrating how they relate to each other. For instance, one side could depict traditional adaptive evolution with natural selection acting on phenotypic diversity, while the other side could illustrate non-adaptive radiation with mutation and DDR at its core.
Author Response
I am grateful for the reviewers comments and appreciation expressed for the complexity of the subject. A simple diagram depicting Non-adaptive radiation vs. neo-Darwinian adaptive radiation has been prepared and will be added.
Round 2
Reviewer 1 Report
Dear Editor,
I am glad to see the author appreciated my comments. I think the manuscript has improved from the first round of revision and it is now ready to be published in Biology.
With kind regards